# Protectivity of COVID-19 Vaccines and Its Relationship with Humoral Immune Response and Vaccination Strategy: A One-Year Cohort Study

**DOI:** 10.3390/vaccines10081177

**Published:** 2022-07-25

**Authors:** Ferdi Tanir, Burak Mete, Hakan Demirhindi, Ertan Kara, Ersin Nazlican, Gülçin Dağlıoğlu, Filiz Kibar, Salih Çetiner, Ceren Kanat

**Affiliations:** 1Department of Public Health, Faculty of Medicine, Çukurova University, Adana 01330, Turkey; ftanir@cu.edu.tr (F.T.); bmete@cu.edu.tr (B.M.); ekara@cu.edu.tr (E.K.); enazlican@cu.edu.tr (E.N.); ckanat@per.cu.edu.tr (C.K.); 2Central Laboratory, Balcalı Hospital, Faculty of Medicine, Çukurova University, Adana 01330, Turkey; gdaglioglu@cu.edu.tr (G.D.); fkibar@cu.edu.tr (F.K.); scetiner@cu.edu.tr (S.Ç.)

**Keywords:** SARS-CoV-2, inactivated vaccine, mRNA vaccine, COVID-19, homologous vaccination, heterologous vaccination, protectivity

## Abstract

This prospective cohort study aimed to evaluate the efficacy of COVID-19 vaccine schemes, homologous versus heterologous vaccine strategies, and vaccine-induced anti-S-RBD-IgG antibody response in preventing COVID-19 among 942 healthcare workers 1 year after vaccination with the inactivated and/or mRNA vaccines. All participants received the first two primary doses of vaccines, 13.6% of them lacked dose 3, 50.5% dose 4, and 90.3% dose 5. Antibody levels increased with the increase in number of vaccine doses and also in heterologous vaccine regimens. In both inactive, mRNA vaccines and mixed vaccination, infection rates were significantly higher in two-dose-receivers, but lower in four- or five-dose receivers and increasing the total number of vaccine doses resulted in more protection against infection: the three-dose regimen yielded 3.67 times more protection, the four-dose 8 times, and five-dose 27.77 times more protection from COVID-19 infection, compared to any two-dose vaccination regimens. Antibody levels at the end of the first year of four- or five-dose-receivers were significantly higher than two- or three-dose receivers. To conclude, an increased number of total vaccine doses and anti-S-RBD antibody levels increased the protection from COVID-19 infection. Therefore, four or more doses are recommended in 1 year for effective protection, especially in risk groups.

## 1. Introduction

COVID-19 (COronaVIrusDisease-19) vaccines emerged as a hope to control and end the pandemic caused by SARS-CoV-2 (severe acute respiratory syndrome-coronavirus-2), they were awarded emergency usage license (EUL) without waiting for clinical trials to be completed. Both logistical issues (like production, global/local delivery, and fair distribution) and scientific questions (like vaccine efficacy, safety, optimization of vaccine regimens, booster dosing, and protection relevance) were the main concern of researchers and decision-makers [1]. The vaccination in Turkey started with the inactivated vaccine CoronaVac^TM^ [2]. In late December 2021, the sharp increase in the number of infected cases all over the World and in Turkey, of course, was related to the emergence of the SARS-CoV-2 variant B.1.1.529 (Omicron), which was more capable of evading the immune system and the immunity of individuals was recessed as more than 4 months had passed since the last shot [3]. Israel, Chile, Denmark, and Turkey were countries that adopted the three-dose strategy over the four-dose strategy. We aimed to determine the effectiveness and protectivity from COVID-19 infection of different COVID-19 vaccines in terms of schedules (two-/three-/four-/five-dose schemes), new strategies (homologous versus heterologous vaccination), and vaccine-induced humoral antibody (anti-S-RBD-IgG) levels in a group of health care workers and the incidence of adverse events at the end of 1-year follow-up.

## 2. Material and Methods

### 2.1. Study Design and Participants

This prospective cohort study was carried out at Cukurova University (Adana, Turkey) between February 2021 and 2022 (1 year follow-up since the initiation of vaccination in health care workers in Turkey) and included health care workers who had been vaccinated with the inactivated SARS-CoV-2 and BNT162b2 mRNA vaccines in the context of a public vaccination program by the Turkish MoH. The minimum sample size was calculated as 945 participants by assuming type-1 error as 0.05, type-2 error as 0.1, and effect size as 0.02 (η_2_ = 0.02, small effect). The participants were randomly selected from a list of 3000 health care workers with substitution lists. A total of 1000 health care workers participated in the first step of the study, decreasing to 942 due to the lack or incompleteness of some results. All participants signed an informed consent form after required information.

### 2.2. Vaccine Information

#### 2.2.1. Inactivated SARS-CoV-2 Vaccine by Sinovac (CoronaVac^TM^)

The vaccine administered to health care workers by the Turkish MoH was the “inactivated SARS-CoV-2 vaccine (CoronaVacTM)”, with aluminium hydroxide, developed by Sinovac Biotech Ltd., Life Sciences Lab., Beijing, China. The vaccine (that will be named shortly as CV) was administered intramuscularly in the deltoid region of the upper arm with a dosage of 3 μg/0.5 mL.

#### 2.2.2. BNT162b2 mRNA Vaccine by Pfizer & BioNTech (Comirnaty^®^)

The vaccine BNT162b2 (Comirnaty^®^) produced by BioNTech Manufacturing GmbH Germany is a nucleoside-modified messenger-RNA (mRNA) encapsulated in lipid nanoparticles (LNP), which enables the delivery of the RNA into host cells to allow expression of the SARS-CoV-2 spike (S) antigen. This vaccine (that will be named shortly as BNT) is a white to off-white frozen suspension provided as a multiple-dose vial and must be diluted before use. One vial (0.45 mL) contains six doses of 0.3 mL after dilution. One dose (0.3 mL) contains 30 micrograms of COVID-19 mRNA vaccine (embedded in lipid nanoparticles) [4].

#### 2.2.3. Mixed (Heterologous) Vaccine Administration

The Turkish Republic MoH declared the introduction of additional third and fourth doses, in June and August 2021, respectively, to be administered to health care workers and the elderly, who had previously received two doses of CV and to the individuals who wished to be vaccinated due to some international travel requirements. All individuals were given the right to choose between CV and BNT vaccines of their free will.

### 2.3. Immun Response Assessments

Our project aimed to determine the seroconversion in the context of anti-SARS-CoV-2 S-RBD (anti-S-RBD) immunoglobulin G (IgG) antibodies in 195 health care workers at 1, 3, and 6 months following the initial two doses of COVID-19 vaccines. In the present fourth step of the project, anti-S-RBD IgG antibodies were measured in 942 participants who completed 12 months after the initial administration of two doses of CV. Among 942 people, 195 belonged to the 1-year follow-up group included in the first three steps of the project (i.e., post-initial two doses at 1st, 3rd and 6th months), while 747 were recruited in the project at the end of the first year. Therefore, their antibody analysis consisted of the measurements of the 12th month. As mentioned in the previous paragraph, different cohorts were formed according to the vaccine type preference (CV and/or BNT) of the individuals:-**The vaccine cohorts-A classified according to the vaccine dosing-scheme subgroups;**-2-dose-CV-receivers-3-dose-CV-receivers-4-dose-CV-receivers-2-dose-BNT-receivers-3-dose-BNT-receivers-2-dose-CV + 1-dose-BNT-receivers-3-dose-CV + 1-dose-BNT-receivers-2-dose-CV + 2-dose-BNT-receivers-2-dose-CV + 3-dose-BNT-receivers-**The vaccine cohorts-B classified according to the vaccine types (homologous or heterologous)**-Homologous CV (only CV-receivers)-Homologous BNT (only BNT-receivers)-Heterologous (both CV and BNT-receivers)

### 2.4. Laboratory Procedure

About 5 mL of blood samples were collected into biochemistry tubes with vacuum gel. The sera were extracted by centrifugation at 3000 g for 10 min and kept at 2–8 °C for 1–3 days. Test calibrators and controls were performed first. After the control results were observed to be within the expected ranges, the samples were tested by trained experts in the accredited (by the Joint Commission International (JCI) since 2006) Central Laboratory of Cukurova University Balcali Hospital, Adana, Turkey with the MAGLUMI 2000 series fully automated chemiluminescence immunoassay analyzer (CLIA) (Snibe Diagnostics, Shenzen New Industries Biomedical Engineering Co. Ltd., Shenzhen, China). The test kit for the determination of antibodies was MAGLUMI^®^ SARS-CoV-2 S-RBD IgG (CLIA) (Cat.#130219017M) (Snibe Diagnostics, Shenzen New Industries Biomedical Engineering Co. Ltd., Shenzhen, China). The SARS-CoV-2 S-RBD IgG (CLIA) assay is an indirect chemiluminescence immunoassay. The analyzer automatically calculates the numerical output in each sample using a calibration curve, which is generated by a two-point calibration master curve procedure. The results are expressed in absorbance units (AU/mL). The results are reported to the end-user as “Reactive” and “Non-Reactive”, where “Non-Reactive” indicates a result less than 1.00 AU/mL (<1.00 AU/mL) and “Reactive” indicates a result greater than or equal to 1.00 AU/mL (≥1.00 AU/mL) [5]. The test is only for use according to the Food and Drug Administration’s Emergency Use Authorization [6]. The SARS-CoV-2 S-RBD IgG test is an indirect CLIA and has a high correlation with VNT50 titres (R = 0.712), where VNT stands for “Virus Neutralization Test”, which is a gold standard for quantifying the titer of neutralizing antibodies (nAbs) for a virus [7].

### 2.5. Statistical Analyses

Data were examined using the SPSS 22 statistical analyses package (2013, IBM, New York, NY, USA). Non-parametric tests were used in the analysis of categorical and non-normally distributed data, while parametric tests were used in the analysis of normally-distributed data. Dunn’s test was used in post-hoc analyses. The enter model was used in the regression analyses and the models were univariate (number of doses in Cox regression, and Anti-S-RBD level in logistic regression). Data were analyzed by Mann–Whitney U, Kruskal–Wallis, Freidman, Chi-Square, Logistic Regression, and Cox Regression test. A value of *p* < 0.05 was considered significant.

## 3. Results

The mean age of 942 participants in the study was 41.17 ± 11.28 (between 17–72). The distribution of the participants according to work positions was 195 physicians (20.7%), 179 nurses (19%), and 568 other positions (60.3%). Reminding that the vaccination in Turkey started on 15 February 2021, 303 (32.2%) participants reported to have been infected with COVID-19 before (199 individuals) or within 1 year (104 individuals) from the start of vaccination. Reinfection was observed in seven participants (five between the second and third doses, one between the third and fourth doses, and one after the fourth dose). Hospitalization was required in 21 patients, of which 18 were infected in the pre-vaccination period, and 3 in the post-vaccination period. At the end of the first year, only six participants had non-reactive antibody levels. The distribution of anti-S-RBD IgG levels of individuals and the rates of non-reactive ones according to demographic characteristics and vaccine cohorts were given in Table 1. It was found that antibody levels increased significantly in correlation with the increase in the number of vaccine doses, and the increase in antibody levels was significantly higher in heterologous vaccine regimens.

All of the participants were administered the initial two doses of vaccines, but 13.6% of them did not receive dose-3, while 50.5% did not receive dose-4, and 90.3% did not receive dose-5. While the interval between dose 1 and 2 was found as a mean of 38 days, that between dose 2 and 3 averaged between 130–169 days, that between dose 3 and 4 averaged between 55–167 days, and that between dose and 4–5 as 128 days. The intervals between doses according to the vaccine schemes and the follow-up times from the time the first vaccine dose was administered in each of the vaccine cohorts were given in Appendix A.

A statistically significant difference was found when the status of being infected with COVID-19 was compared by vaccine scheme subgroups (vaccine cohorts-A). The rate of being infected with COVID-19 was found to be significantly higher both in two-dose-CV-receivers and in two-dose-BNT-receivers. In contrast, the rate of being infected with COVID-19 was found to be significantly lower in two-dose-CV+two-dose-BNT-receivers and in two-dose-CV+three-dose-BNT-receivers.

When the infection rates were compared by total vaccine doses, infection rates were found significantly higher in two-dose-receivers, but lower in those who received four or five doses of vaccines. No difference in infection rates was observed in three-dose-receivers (Table 2).

The Cox regression model formulated to estimate the risk of being infected with COVID-19 based on the total number of vaccine doses, regardless of vaccine types, was found to be predictive. It was found that the risk of infection decreased as the number of doses increased in all three vaccine cohorts (Figure 1). In the model, vaccine doses were stratified by vaccine cohort B. The dependent variable of the model was “being infected with COVID-19”, and the independent variable was the total number of vaccine doses (with reference = 2-dose-receivers). The increase in the number of doses was found to be more protective against COVID-19 infection. Compared to two-dose administration, three-dose administration was found to be 3.67 times more protective from the infection (H.R. (hazard ratio) = 0.272), 8 times (H.R. = 0.125) more protective in case of four-dose, and 27.77 times (H.R. = 0.036) more protective in case of five-dose (Table 3).

The logistic regression model, including participants infected in the post-vaccination period, established to predict the effect of anti-S-RBD-IgG levels (independent) on protection from COVID-19 infection (dependent), which was shown to be significant (*p* < 0.001). Each 0.008-unit increase in the anti-S-RBD-IgG levels was observed to increase the protectivity from being infected with COVID-19 by 1.008-fold with an odds ratio of 0.992 (95% confidence interval between 0.989–0.996).

Regardless of the vaccine type, in months 1, 3, 6, and 12, anti-S-RBD-IgG levels were compared between (inter) and within (intra) vaccine-dose subgroups. After month 6, intragroup antibody levels continued to increase in the four- or five-dose-receivers, but decreased in two- or three-dose-receivers. At the end of year-1, inter-group antibody levels were found to be higher in four- or five-dose-receivers than two- or three-dose-receivers (Table 4).

In our study, 35.9% of all participants did not declare any adverse event. The most common adverse events observed after any of the doses were pain at the injection site, malaise, fatigue, myalgia, backache, and fever. The rate of adverse events was observed to increase after dose 3, but no serious events were detected. The table of adverse events was presented as Appendix A.

## 4. Discussion

The key to controlling the COVID-19 pandemic is vaccinating the entire population at full schedule including boosters. The success of this policy is hampered by the occurrence of infection and disease in fully vaccinated persons. The potential primary cause of infection despite vaccination is the emergence of new variants that evade immunity, thereby reducing the efficacy of the vaccine. Another potential cause of infection is a decrease in the immunity provided by the vaccine or disease itself because of time or other factors [8].

To start with the immunity, regardless of vaccine type, we found a continuing increase of antibody levels after the month 6 in four-/five-dose-receivers, but a decrease in two-/three-dose-receivers. At the end of year 1, this difference was still significant. Similarly to our findings, following BNT-dose-2, Mizrahi et al. [9], Puranik et al. [10], and Khoury et al. [11] reported a decrease in vaccine-derived neutralizing antibody titres at month 6; Goldberg et al. [8] in all age groups after a few months; Levin et al. [12] in male, immunosuppressed, and 65-years-old and over individuals at month 6; and Thomas et al. (in a longer follow-up of phase 2–3 randomized trial of BNT) [13] a 96–84% reduction in vaccine efficacy between month 4 and 7. Regarding CV, Demirhindi et al. [14] reported a 60% decrease in indirect neutralizing antibody concentrations at month 6 compared to month 3 in two-dose-CV-receivers, but a 5–20 times increase in three-dose-receivers (CV and/or BNT).

Obviously, increased antibody responses or serostability point out efficacy in terms of humoral immunity, but this does not guarantee protectivity. One year of usage and follow-up gave us chance to evaluate the protectivity of the vaccines from the COVID-19 infection, besides vaccine efficacy.

We evaluated the relationship between protectivity and vaccination schedule and found the number of vaccine doses to be inversely proportional to infection rates regardless of vaccine type: 32.6% of infection rate in two-dose-receivers, 16.0% in three-dose-receivers, 8.8% in four-dose-receivers, and 4.0% in five-dose-receivers. Regardless of the vaccine type, we found that three-dose-receivers were protected approximately 3.67 times more, four-dose-receivers 8 times more, and five-dose-receivers 27.77 times more from being infected with COVID-19 than any two doses of any COVID-19 vaccine type. Similar proportionality was observed by other researchers. Spitzer et al. reported an incidence rate of infection of 12.8 per 100,000 person-days in three-dose-BNT-receivers; in contrast to 116 in unvaccinated individuals [15], while Bar-On et al., found it as 1.5 in four-dose-BNT-receivers, 3.9 in three-dose-BNT-receivers in the case of severe disease, and 4.2 in the control group. At week 4, BNT-dose-4 reported a lower rate of confirmed infection than three-dose-BNT-receivers by a factor of 2.0 (by a factor of 3.5 in severe infection) compared to a factor of 1.8 observed in the control group (by a factor of 2.3 in severe infection). The protection was reported to wane in the following weeks, but not in severe infection for at least 6 weeks after dose 4. Magen et al., revealed that BNT-dose-4 was effective in reducing the short-term risk of COVID-19-related outcomes in people who, at least 4 months ago, had received BNT-dose-3 [16]. On days 7–30 after dose 4, the efficacy of the vaccine was estimated as 45% against SARS-CoV-2 infection confirmed by polymerase chain reaction, 55% against symptomatic COVID-19, 68% against COVID-19-related hospitalization, 62% against severe COVID-19, and 74% against COVID-19-related death. On days 7–30 after the BNT-dose-4, the absolute risk difference for COVID-19-related hospitalization (BNT-dose-3 versus BNT-dose-4) was found as 180.1 cases per 100,000 and 68.8 for severe COVID-19 [17].

When we evaluated the odds of being infected with COVID-19 as a function of vaccine-induced antibody levels, the protectivity could be expressed as: every unit increase of 0.008 in the antibody concentration resulted in a 1.008 times (Odds Ratio = 0.992) decrease in the infection risk, and with the decrease of the antibody levels over time, the effectiveness of prevention from COVID-19 also decreased. We calculated the hazard ratio (HR) as 0.272 for dose 3 regardless of vaccine type, Spitzer et al. reported HR as 0.07 [15]. At least at day 12, Bar-On et al., reported the confirmed infection rate to be 11.3 times lower in the BNT-dose-3-receivers (19.5 times in severe disease) compared to the no-booster group and 5.4 times lower than the rate observed on days 4–6 [1].

At this point, even though increased efficacy of booster doses for protection from severe COVID-19 and reduced risk of contagion are evident, uncertainties regarding the efficacy and safety of vaccines cause a decrease in the motivation of the population to take a booster dose. Hesitancies are generally due to the adverse events encountered in previous vaccination schedules, thinking that the booster dose was administered too early and uncertainty about the increased efficacy caused by booster doses [17]. In our study, 35.9% of all participants did not declare any adverse event. The most common adverse events observed after any of the doses were pain at the injection site, malaise, fatigue, myalgia, backache, and fever. After dose 3, the rate of adverse events seemed to increase somewhat, but no serious events were detected.

The limitations of the study: (1) The study group consisting of only healthcare professionals, (2) a relatively small sampling, (3) lower participant numbers in some subgroup analyses, and (4) lack of the analyses about protectivity from severe disease due to this low numbers.

The strengths of the study: (1) It is one of the few studies that evaluate the protection from COVID-19 infection concerning four or five vaccine doses, regardless of the vaccine type; (2) it evaluates the effect of antibody levels on the protection; (3) it includes long-term results (i.e., one year); and (4) it is one of the few studies examining the heterologous administration of an inactivated with an mRNA vaccine.

## 5. Conclusions

Higher antibody levels and administration of four and/or five doses of vaccines are more protective from COVID-19 than 2 or 3 doses. The same is true in the heterologous vaccination strategy with a stronger antibody response observed in cohorts containing BNT vaccine. High-risk groups like healthcare workers, the elderly, and immunocompromised individuals are recommended at least four doses of vaccines regardless of the vaccine type, with a resulting “0-1-5-9-months scheme” in 1 year.

## Figures and Tables

**Figure 1 vaccines-10-01177-f001:**
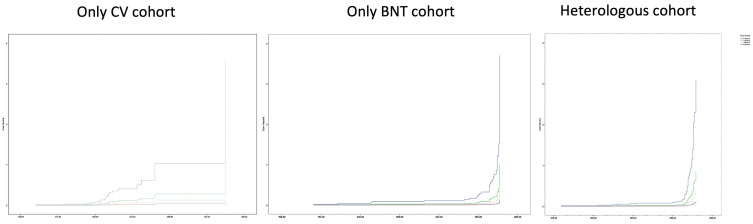
Hazard function by vaccine cohort and number of doses.

**Table 1 vaccines-10-01177-t001:** Anti-S-RBD levels at month-12 by sociodemographic characteristics and vaccine cohorts.

Sociodemographic Characteristics	All n (%)	n of Negative COVID-19 History ^1^	Anti-S-RBD IgG (AU/mL) ^1^ [Median (IQR)]	*p*-Value ^2^	NR (%) ^3^
**Sex**					
Male	404 (42.9)	282	169.50 (98.82)	0.028 *	1.1
Female	538 (57.1)	357	186.40 (94.25)		0.8
**Age**					
15–29	186 (19.7)	121	175.90 (104.00)	0.693	0.0
30–44	372 (39.5)	234	178.00 (102.48)		1.7
45–59	321 (34.1)	234	181.40 (84.20)		0.9
60 and older	63 (6.7)	50	182.00 (87.75)		0.0
**Chronic comorbidity**					
Yes	288 (30.6)	209	169.60 (93.55)	0.092	0.5
No	654 (69.4)	430	186.60 (98.67)		1.2
**Total vaccination doses received**					
2-dose-receivers	128 (13.6)	63	93.18 (154.46)	<0.001 *	6.3
3-dose-receivers	348 (36.9)	225	141.90 (98.15)		0.9
4-dose-receivers	374 (39.7)	279	191.60 (79.00)		0.0
5-dose-receivers	92 (9.8)	72	208.00 (49.98)		0.0
**Vaccine schedule cohorts-A ^3^**					
2-dose-CV	45 (4.8)	23	4.13 (81.79)	<0.001 *	17.4
3-dose-CV	53 (5.6)	33	11.98 (62.63)		3.0
4-dose-CV	15 (1.6)	9	30.22 (118.36)		0.0
2-dose-BNT	84 (8.9)	41	118.30 (151.07)		0.0
3-dose-BNT	94 (10.0)	62	183.05 (80.27)		0.0
2-dose-CV + 1-dose-BNT	200 (21.2)	129	141.80 (92.65)		0.8
3-dose-CV + 1-dose-BNT	11 (1.2)	8	241.75 (74.17)		0.0
2-dose-CV + 2-dose-BNT	349 (37.0)	263	195.70 (76.40)		0.0
2-dose-CV + 3-dose-BNT	91 (9.7)	71	207.60 (50.00)		0.0
**Vaccine cohorts-B ^4^**					
Homologous CV	113 (12.0)	65	12.29 (79.56)	<0.001 *	7.7
Homologous BNT	178 (18.9)	103	170.40 (99.10)		0.0
Heterologous	651 (69.1)	471	189.30 (86.50)		0.2
Total	942 (100.0)	639			

CV: CoronaVac^TM^ BNT:Comirnaty^®^. ^1^ Those infected with COVID-19 (n = 303) were excluded. ^2^ AFTER ADDING * FOOTER (2) CHANGED *p* values in the comparison of the antibody levels. ^3^ Row percentage of NR (non-reactive) referring to those with antibody levels <1 AU/mL. Subgroups by ^3^ vaccine dosing schemes and ^4^ vaccine types (homologous or heterologous). * Significant *p*-values.

**Table 2 vaccines-10-01177-t002:** Comparison of COVID-19 infection history by vaccine schedules and number of doses.

Vaccine Cohorts-A	COVID-19 Infection History ^1^		*p*-Value
	Yes (%)	No (%)	
2-dose-CV *	14 (37.8)	23 (62.2)	<0.001
3-dose-CV	9 (21.4)	33 (78.6)	
4-dose-CV *	3 (25.0)	9 (75.0)	
2-dose-BNT *	17 (29.8)	40 (70.2)	
3-dose-BNT	5 (7.5)	62 (92.5)	
2-dose-CV + 1-dose-BNT	29 (18.4)	129 (81.6)	
3-dose-CV + 1-dose-BNT	1 (11.1)	8 (88.9)	
2-dose-CV + 2-dose-BNT *	23 (8.0)	263 (92.0)	
2-dose-CV + 3-dose-BNT*	3 (4.1)	71 (95.9)	
**Vaccine cohorts-B**			
Homologous CV *	26 (28.6)	65 (71.4)	<0.001
Homologous BNT	22 (17.7)	102 (82.3)	
Heterologous *	56 (10.6)	471 (89.4)	
**Total doses received**			
2-dose-receivers *	31 (33.3)	62 (66.7)	<0.001
3-dose-receivers	43 (16.0)	225 (84.0)	
4-dose-receivers *	27 (8.8)	279 (91.2)	
5-dose-receivers *	3 (4.0)	72 (96.0)	

^1^ Participants who were infected after being vaccinated (n = 104) were included in the comparison of COVID-19 infection history rates. * Denotes the cohorts where the COVID-19 infection history rates were significantly different.

**Table 3 vaccines-10-01177-t003:** Protectivity from COVID-19 infection by the number of vaccine doses.

	B	*p*	H.R.	95% CI for H.R.	
				Lower Limit	Upper Limit
2-dose		<0.001			
3-dose	−1.301	<0.001	0.272	0.148	0.501
4-dose	−2.080	<0.001	0.125	0.058	0.269
5-dose	−3.314	<0.001	0.036	0.010	0.138

Cox Regression analysis: Only those infected in the post-vaccination period were added to the model (104 individuals), while those infected in the pre-vaccination period were excluded.

**Table 4 vaccines-10-01177-t004:** Change in the antibody levels over time according to the vaccine-dose subgroups.

Dose Subgroups	Anti-S-RBD-IgG Levels Median (IQR) ^a^				Intra-Group p ^b^
	Month-1	Month-3	Month-6	Month-12	
2-dose-receivers (n = 7)	32.72 (91.49)	39.82 (72.63)	4.34 (37.17)	0.91 (23.49)	0.002 *
3-dose-receivers (n = 44)	33.06 (66.82)	9.23 (15.68)	133.60 (44.04)	118.95 (50.59)	<0.001 *
4-dose-receivers (n = 77)	24.84 (53.91)	9.20 (16.16)	137.50 (12.75)	189.40 (95.75)	<0.001 *
5-dose-receivers (n = 13)	14.69 (46.74)	6.28 (21.71)	135.95 (7.83)	204.60 (41.35)	<0.001 *
**inter-group p** ^b^	0.720	0.511	0.004 *	<0.001 *	

* Statistically significant differences. The analyses belonged to 141 people who did not have COVID-19 in the sub-groups (n = 195). ^a^ IQR = Inter-quartile ranges. ^b^ Comparison within (intra) dose-subgroups.

## Data Availability

The data presented in this study are available on request from the corresponding author. The data are not publicly available due to personal data protection regulations.

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
