# Peer review of "Protectivity of COVID-19 Vaccines and Its Relationship with Humoral Immune Response and Vaccination Strategy: A One-Year Cohort Study"

_vaccines, 2022, doi:10.3390/vaccines10081177_

Round 1
Reviewer 1 Report
This interesting prospective cohort study measures the efficacy of the vaccination against COVID-19 and the follow-up of the immune response.
The introduction is clear and presents the rationale. However, there are two vaccines used in Turkey (CoronaVac and Comirnaty) but references about CoronaVac are missing. It would be interesting to add references regarding its effectiveness.
In the materials and methods section, the population, location, study inclusion period, follow-up, vaccine information and laboratory procedure are well defined. The data collected are listed but the statistical analyses are described too briefly. At a minimum, it is necessary to detail whether post-hoc tests were performed, how the models were made, and whether the diagnostic criteria of these models were checked.
In the results section, the demographic characteristics and the different cohorts are well described. In table 1, it is difficult to know to which comparison the p-values correspond. It would be appropriate to make this clear. In lines 174-175, the mean of interval between the dose-1-2 is alone while all groups had at least two doses. In lines 185 to 192, the sentences are not easy to understand because if the comparisons are about the rate of being infected with COVID-19, the message is blurred by the presence of time delays between doses.
In table 3, it is also difficult to know to which comparison the p-values correspond. The presence of symbols “a” and “b” does not make it possible to understand which group differs from the others. For the Cox model, it is not clear why only the number of doses was used in the model when the type of vaccine clearly has an effect. It would be relevant to introduce this variable in the model and to check its goodness-of-fit.
In line 216, why did you take an increase of 0.008-unit in the anti-S-RBD-IgG levels instead of 1 unit? And to analyze the anti-S-RBD-IgG levels over time, a mixed model could have been used to take into account the time.
Finally, the incidence of adverse events that appears in the objectives does not appear in the results section.
The discussion part is clear and well written but the references are not indicated in the same way. In line 257, it seems that "2-dose-receivers" should be replaced by "3-dose-receivers". In lines 264 to 266, it is not clear which group has 1.8 times less infection than the control group.
In the conclusion, the difference of efficacy between the two vaccines is missing.
Author Response
The author's reply to the Review Report (Reviewer 1) at Round 1 has been uploaded as a pdf file. Please see the attachment.

Reviewer 2 Report
This is an interesting study concerning protectivity of Covid-19 vaccines in a cohort of healthcare workers one-year after the initial vaccination.
The title should be changed to humoral immune response. The introduction and the methods are adequately presented. There some important points to be addressed in the results section and in the discussion.
In table 1 vaccine schedule cohorts - A, statistical significance for BNT-3 dose etc is not presented. Table 2 could be presented in supplementary material, too many details not very interesting. Paragraph lines 184-192 is not clear. The time the blood sample was taken is irrelevant. The comparisons and mean interval days are not clear.
On line 195 no difference with what?
In table 3 all significant comparisons had the same significance p<0,001?
In table 5 what do the parentheses numbers mean? Why in 2 dose recipients the levels are higher at 3 months than at 1 month and different than the rest of the groups? All the other groups show a decrease at 3 months.
In the discussion there is extensive comparison with the references which is very good but difficult to follow reading the text. Perhaps a table with the results of the references could help regarding paragraphs lines 236-248 and 253-276.
Author Response
The author's reply to the Review Report (Reviewer 2) at Round 1 has been uploaded as a pdf file. Please see the attachment.

Reviewer 3 Report
This is no prospective study, if the participants are recruited in februrary 2022. The most important data are missing or not declared:
1) 303 of 942 participants are excluded of cause COVID-19 infection. We need the re infection rate of this group to estimate natural immunity
2) 104 of 639 participants had breakthrough infection despite at least a complete basic immunization. Based upon the registration study Biontec propose efficacy of >97% after basic immunization. Curevac vaccine was canceled of cause 53% efficacy. The 70,2% efficacy of BNT in the current study after basic immunization is only slightly better.
3) The side effects with the increasing number of vaccination was only declared in the discussion ("after dose-3 the rate of adverse events seemed to increase somewhat...."). This is very poor!
4) In the discussion the authors reported efficacy on days 7 to 30 after dose -3/4/5. This is completely not interesting in a group with a mean age of 41 years, most without comorbidities). In this group vaccination have to protect at least one season between nomvember to april (6month).
5) The very low titers of antibodies in homologous CV should be discussed. Is it really a vaccination?
Author Response
The author's reply to the Review Report (Reviewer 3) at Round 1 has been uploaded as a pdf file. Please see the attachment.

Round 2
Reviewer 1 Report
Overall, the answers provided are satisfactory. Nevertheless, some answers could have been improved, particularly if the proportionnal hazards assumption had been checked for the Cox model. I think that the introduction of the type of vaccine in the model would increase the relevant of the Cox model even if the cohorts were heterologous.
Finally, I think that this sentence "When the time interval between the time when the dose-1 was received and the time when the blood samples were taken" could be deleted.
Author Response
Criticism recommendation and authors’ answers
- Overall, the answers provided are satisfactory. Nevertheless, some answers could have been improved, particularly if the proportionnal hazards assumption had been checked for the Cox model. I think that the introduction of the type of vaccine in the model would increase the relevant of the Cox model even if the cohorts were heterologous.
Reply 1:
Dear referee
It is certain that the type of vaccine has something to do with protection. However, since there were no individuals who were never vaccinated, the situation of which cohort should be referenced in the establishment of the model is mixed. In line with your recommendation, vaccine types are included in the model. Total dose cohorts were stratified by vaccine type cohorts (cohort B). New analyzes results have been added in line 203-213.
- Finally, I think that this sentence "When the time interval between the time when the dose-1 was received and the time when the blood samples were taken" could be deleted.
Reply 2: The sentence has been deleted. (Line 187-189).
Not: As a result of the new analyzes, the places changed in the text are colored yellow.

Reviewer 3 Report
Thank you for the revisions
Author Response
Thank you...